# Neural Machine Translation via Binary Code Prediction

## Abstract

In this paper, we propose a new method for calculating the output layer in neural machine translation systems. The method is based on predicting a binary code for each word, and can reduce computation time/memory requirements of the output layer to be logarithmic in vocabulary size in the best case. In addition, we also introduce two advanced approaches to improve robustness of the proposed model: using error-correcting codes and combining softmax and binary codes. Experiments show the proposed model achieves translation accuracies that approach the softmax, while reducing memory usage on the order of 1/10 to 1/1000, and also improving decoding speed on CPUs by x5 to x20.

## 1 Introduction

When handling broad or open domains, machine translation systems usually have to handle a large vocabulary as their inputs and outputs. This is particularly a problem in neural machine translation (NMT) models (Sutskever et al., 2014), such as the attention-based model (Bahdanau et al., 2014; Luong et al., 2015) shown in Figure 1. In these models, the output layer is required to generate a specific word from an internal vector, and a large vocabulary size tends to require a large amount of computation to predict each of the candidate word probabilities.

Because this is a significant problem for neural language and translation models, there are a number of methods proposed to resolve this problem, which we detail in Section 2.2. However, none of these previous methods simultaneously satisfies the following desiderata, all of which, we argue, are desirable for practical use in NMT systems:

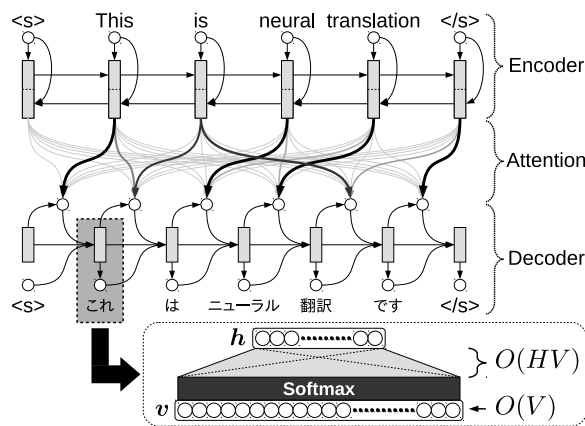

Figure 1: Encoder-decoder-attention NMT model and computation amount of the output layer.

**Memory efficiency:** The method should not require large memory to store the parameters and calculated vectors to maintain scalability in resource-constrained environments.

**Time efficiency:** The method should be able to train the parameters efficiently, and possible to perform decoding efficiently with choosing the candidate words from the full probability distribution. In particular, the method should be performed fast on general CPUs to suppress physical costs of computational resources for actual production systems.

**Compatibility with parallel computation:** It should be easy for the method to be mini-batched and optimized to run efficiently on GPUs, which are essential for training large NMT models.

In this paper, we propose a method that satisfies all of these conditions: requires significantly less memory, fast, and is easy to implement mini-batched on GPUs. The method works by not pre-

dicting a softmax over the entire output vocabulary, but instead by encoding each vocabulary word as a vector of binary variables, then independently predicting the bits of this binary representation. In order to represent a vocabulary size of $2^n$, the binary representation need only be at least $n$ bits long, and thus the amount of computation and size of parameters required to select an output word is only $O(\log V)$ in the size of the vocabulary $V$, a great reduction from the standard linear increase of $O(V)$ seen in the original softmax.

While this idea is simple and intuitive, we found that it alone was not enough to achieve competitive accuracy with real NMT models. Thus we make two improvements: First, we propose a hybrid model, where the high frequency words are predicted by a standard softmax, and low frequency words are predicted by the proposed binary codes separately. Second, we propose the use of convolutional error correcting codes with Viterbi decoding (Viterbi, 1967), which add redundancy to the binary representation, and even in the face of localized mistakes in the calculation of the representation, are able to recover the correct word.

In experiments on two translation tasks, we find that the proposed hybrid method with error correction is able to achieve results that are competitive with standard softmax-based models while reducing the output layer to a fraction of its original size.

## 2 Problem Description and Prior Work

### 2.1 Formulation and Standard Softmax

Most of current NMT models use *one-hot representations* to represent the words in the output vocabulary – each word $w$ is represented by a unique sparse vector $\boldsymbol{e}_{\mathrm{id}(w)} \in \mathbb{R}^V$, in which only one element at the position corresponding to the word ID $\mathrm{id}(w) \in \{x \in \mathbb{N} \mid 1 \leq x \leq V\}$ is 1, while others are 0. $V$ represents the vocabulary size of the target language. NMT models optimize network parameters by treating the one-hot representation $\boldsymbol{e}_{\mathrm{id}(w)}$ as the true probability distribution, and minimizing the cross entropy between it and the softmax probability $\boldsymbol{v}$:

$$L_{\mathcal{H}}(\boldsymbol{v}, \mathrm{id}(w)) \;:=\; \mathcal{H}(\boldsymbol{e}_{\mathrm{id}(w)}, \boldsymbol{v}), \qquad (1)$$
$$=\; \log \mathrm{sum}\exp \boldsymbol{u} - u_{\mathrm{id}(w)}, \quad (2)$$
$$\boldsymbol{v} \;:=\; \exp \boldsymbol{u} / \mathrm{sum}\exp \boldsymbol{u}, \qquad (3)$$
$$\boldsymbol{u} \;:=\; W_{hu}\boldsymbol{h} + \boldsymbol{\beta}_u, \qquad\qquad (4)$$

where $\mathrm{sum}\,\boldsymbol{x}$ represents the sum of all elements in $\boldsymbol{x}$, $x_i$ represents the $i$-th element of $\boldsymbol{x}$, $W_{hu} \in$

$\mathbb{R}^{V \times H}$ and $\boldsymbol{\beta}_u \in \mathbb{R}^V$ are trainable parameters and $H$ is the total size of hidden layers directly connected to the output layer.

According to Equation (4), this model clearly requires time/space computation in proportion to $O(HV)$, and the actual load of the computation of the output layer is directly affected by the size of vocabulary $V$, which is typically set around tens of thousands (Sutskever et al., 2014).

### 2.2 Prior Work on Suppressing Complexity of NMT Models

Several previous works have proposed methods to reduce computation in the output layer. The *hierarchical softmax* (Morin and Bengio, 2005) predicts each word based on binary decision and reduces computation time to $O(H \log V)$. However, this method still requires $O(HV)$ space for the parameters, and requires calculation much more complicated than the standard softmax, particularly at test time. The *differentiated softmax* (Chen et al., 2016) divides words into clusters, and predicts words using separate part of the hidden layer for each word clusters. This method make the conversion matrix of the output layer sparser than a fully-connected softmax, and can reduce time/space computation amount by ignoring zero part of the matrix. However, this method restricts the usage of hidden layer, and the size of the matrix is still in proportion to $V$.

Sampling-based approximations (Mnih and Teh, 2012; Mikolov et al., 2013) to the denominator of the softmax have also been proposed to reduce calculation at training. However, these methods are basically not able to be applied at test time, still require heavy computation like the standard softmax.

Other methods using characters (Ling et al., 2015) or subwords (Sennrich et al., 2016; Chitnis and DeNero, 2015) can be applied to suppress the vocabulary size, but these methods also make for longer sequences, and thus are not a direct solution to problems of computational efficiency.

## 3 Binary Code Prediction Models

### 3.1 Representing Words using Bit Arrays

Figure 2(a) shows the conventional softmax prediction, and Figure 2(b) shows the binary code prediction model proposed in this study. Unlike the conventional softmax, the proposed method predicts each output word indirectly using dense

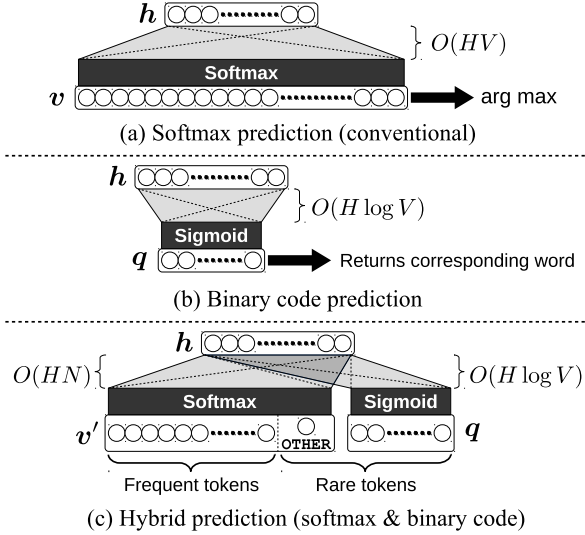

Figure 2: Designs of output layers.

bit arrays that correspond to each word. Let $\boldsymbol{b}(w) := [b_1(w), b_2(w), \cdots, b_B(w)] \in \{0,1\}^B$ be the target bit array obtained for word $w$, where each $b_i(w) \in \{0,1\}$ is an independent binary function given $w$, and $B$ is the number of bits in whole array. For convenience, we introduce some constraints on $\boldsymbol{b}$. First, a word $w$ is mapped to only one bit array $\boldsymbol{b}(w)$. Second, all unique words can be discriminated by $\boldsymbol{b}$, i.e., all bit arrays satisfy that:

$$\mathrm{id}(w) \neq \mathrm{id}(w') \Rightarrow \boldsymbol{b}(w) \neq \boldsymbol{b}(w'). \quad (5)$$

Third, multiple bit arrays can be mapped to the same word as described in Section 3.5. By considering second constraint, we can also constrain $B \geq \lceil \log_2 V \rceil$, because $\boldsymbol{b}$ should have at least $V$ unique representations to distinguish each word. The output layer of the network independently predicts $B$ probability values $\boldsymbol{q} := [q_1(\boldsymbol{h}), q_2(\boldsymbol{h}), \cdots, q_B(\boldsymbol{h})] \in [0,1]^B$ using the current hidden values $\boldsymbol{h}$ by logistic regressions:

$$\boldsymbol{q}(\boldsymbol{h}) = \sigma(W_{hq}\boldsymbol{h} + \boldsymbol{\beta}_q), \quad (6)$$
$$\sigma(\boldsymbol{x}) := 1/(1 + \exp(-\boldsymbol{x})), \quad (7)$$

where $W_{hq} \in \mathbb{R}^{B \times H}$ and $\boldsymbol{\beta}_q \in \mathbb{R}^B$ are trainable parameters. When we assume that each $q_i$ is the probability that "the $i$-th bit becomes 1," the joint probability of generating word $w$ can be represented as:

$$\mathrm{Pr}(w|\boldsymbol{h}) := \prod_{i=1}^{B} \left( b_i q_i + \bar{b}_i \bar{q}_i \right), \quad (8)$$

**Algorithm 1** Mapping words to bit arrays.

**Require:** $w \in \mathcal{V}$
**Ensure:** $\boldsymbol{b} \in \{0,1\}^B$ = Bit array representing $w$

$$x := \begin{cases} 0, & \text{if } w = \text{UNK} \\ 1, & \text{if } w = \text{BOS} \\ 2, & \text{if } w = \text{EOS} \\ 2 + \mathrm{rank}(w), & \text{otherwise} \end{cases}$$
$$b_i := \lfloor x/2^{i-1} \rfloor \mod 2$$
$$\boldsymbol{b} \leftarrow [b_1, b_2, \cdots, b_B]$$

where $\bar{x} := 1 - x$. We can easily obtain the maximum-probability bit array from $\boldsymbol{q}$ by simply assuming $b_i = 1$ if $q_i \geq 1/2$, and $b_i = 0$ otherwise. However, this calculation may generate invalid bit arrays which do not correspond to actual words according to the mapping between words and bit arrays. For now, we simply assume that $w = \text{UNK}$ (*unknown*) when such bit arrays are obtained, and discuss alternatives later in Section 3.5.

The constraints described here are very general requirements for bit arrays, which still allows us to choose between a wide variety of mapping functions. However, designing the most appropriate mapping method for NMT models is not a trivial problem. In this study, we use a simple mapping method described in Algorithm 1, which was empirically effective in preliminary experiments.[1] Here, $\mathcal{V}$ is the set of $V$ target words including 3 extra markers: UNK, BOS (*begin-of-sentence*), and EOS (*end-of-sentence*), and $\mathrm{rank}(w) \in \mathbb{N}_{>0}$ is the rank of the word according to their frequencies in the training corpus. Algorithm 1 is one of the minimal mapping methods (i.e., satisfying $B = \lceil \log_2 V \rceil$), and generated bit arrays have the characteristics that their *higher* bits roughly represents the frequency of corresponding words (e.g., if $w$ is frequently appeared in the training corpus, higher bits in $\boldsymbol{b}(w)$ tend to become 0).

### 3.2 Loss Functions

For learning correct binary representations, we can use any loss function that satisfies a constraint that:

$$L_{\mathcal{B}}(\boldsymbol{q}, \boldsymbol{b}) \begin{cases} = 0, & \text{if } \boldsymbol{q} = \boldsymbol{b}, \\ \geq 0, & \text{otherwise.} \end{cases} \quad (9)$$

For example, the *squared-distance*:

$$L_{\mathcal{B}}(\boldsymbol{q}, \boldsymbol{b}) := \sum_{i=1}^{B} (q_i - b_i)^2, \quad (10)$$

---

[1] Other methods examined included random codes, Huffman codes (Huffman, 1952) and Brown clustering (Brown et al., 1992) with zero-padding to adjust code lengths, and some original allocation methods based on the word2vec embeddings (Mikolov et al., 2013).

or the *cross-entropy*:

$$L_{\mathcal{B}}(\boldsymbol{q}, \boldsymbol{b}) := -\sum_{i=1}^{B} \left(b_i \log q_i + \bar{b}_i \log \bar{q}_i\right), \quad (11)$$

are candidates for the loss function. We also examined both loss functions in the preliminary experiments, and in this paper, we only used the squared-distance function (Equation (10)), because this function achieved higher translation accuracies than Equation (11).

### 3.3 Efficiency of the Binary Code Prediction

The computational complexity for the parameters $W_{hq}$ and $\boldsymbol{\beta}_q$ is $O(HB)$. This is equal to $O(H \log V)$ when using a minimal mapping method like that shown in Algorithm 1, and is significantly smaller than $O(HV)$ when using standard softmax prediction. For example, if we chose $V = 65536 = 2^{16}$ and use Algorithm 1's mapping method, then $B = 16$ and total amount of computation in the output layer could be suppressed to 1/4096 of its original size.

On a different note, the binary code prediction model proposed in this study shares some ideas with the hierarchical softmax (Morin and Bengio, 2005) approach. Actually, when we used a binary-tree based mapping function for $\boldsymbol{b}$, our model can be interpreted as the hierarchical softmax with two strong constraints for guaranteeing independence between all bits: all nodes in the same level of the hierarchy share their parameters, and all levels of the hierarchy are predicted independently of each other. By these constraints, all bits in $\boldsymbol{b}$ can be calculated in parallel. This is particularly important because it makes the model conducive to being calculated on GPGPUs.

However, the binary code prediction model also introduces problems of robustness due to these strong constraints. As the experimental results show, the simplest prediction model which directly maps words into bit arrays seriously decreases translation quality. In Sections 3.4 and 3.5, we introduce two additional techniques to prevent reductions of translation quality and improve robustness of the binary code prediction model.

### 3.4 Hybrid Softmax/Binary Model

According to the Zipf's law (Zipf, 1949), the distribution of word appearances in an actual corpus is biased to a small subset of the vocabulary. As a result, the proposed model mostly learns characteristics for frequent words and cannot obtain enough opportunities to learn for rare words. To alleviate this problem, we introduce a hybrid model using both softmax prediction and binary code prediction as shown in Figure 2(c). In this model, the output layer calculates a standard softmax for the $N - 1$ most frequent words and an OTHER marker which indicates all rare words. When the softmax layer predicts OTHER, then the binary code layer is used to predict the representation of rare words. In this case, the actual probability of generating a particular word is separated into two equations according to the frequency of words:

$$\Pr(w|\boldsymbol{h}) := \begin{cases} v'_{\text{id}(w)}, & \text{if } \text{id}(w) < N, \\ v'_N \cdot \pi(w, \boldsymbol{h}), & \text{otherwise}, \end{cases} \quad (12)$$

$$\boldsymbol{v}' := \exp \boldsymbol{u}' / \operatorname{sum} \exp \boldsymbol{u}', \quad (13)$$

$$\boldsymbol{u}' := W_{hu'}\boldsymbol{h} + \boldsymbol{\beta}_{u'}, \quad (14)$$

$$\pi(w, \boldsymbol{h}) := \prod_{i=1}^{B} \left(b_i q_i + \bar{b}_i \bar{q}_i\right), \quad (15)$$

where $W_{hu'} \in \mathbb{R}^{N \times H}$ and $\boldsymbol{\beta}_{u'} \in \mathbb{R}^N$ are trainable parameters, and $\text{id}(w)$ assumes that the value corresponds to the rank of frequency of each word. We also define the loss function for the hybrid model using both softmax and binary code losses:

$$L := \begin{cases} l_{\mathcal{H}}(\text{id}(w)), & \text{if } \text{id}(w) < N, \\ l_{\mathcal{H}}(N) + l_{\mathcal{B}}, & \text{otherwise}, \end{cases} \quad (16)$$

$$l_{\mathcal{H}}(i) := \lambda_{\mathcal{H}} L_{\mathcal{H}}(\boldsymbol{v}', i), \quad (17)$$

$$l_{\mathcal{B}} := \lambda_{\mathcal{B}} L_{\mathcal{B}}(\boldsymbol{q}, \boldsymbol{b}), \quad (18)$$

where $\lambda_{\mathcal{H}}$ and $\lambda_{\mathcal{B}}$ are hyper-parameters to determine strength of both softmax/binary code losses. These also can be adjusted according to the training data, but in this study, we only used $\lambda_{\mathcal{H}} = \lambda_{\mathcal{B}} = 1$ for simplicity.

The computational complexity of the hybrid model is $O(H(N + \log V))$, which is larger than the original binary code model $O(H \log V)$. However, $N$ can be chosen as $N \ll V$ because the softmax prediction is only required for a few frequent words. As a result, we can control the actual computation for the hybrid model to be much smaller than the standard softmax complexity $O(HV)$,

The idea of separated prediction of frequent words and rare words comes from the differentiated softmax (Chen et al., 2016) approach. However, our output layer can be configured as a fully-connected network, unlike the differentiated soft-

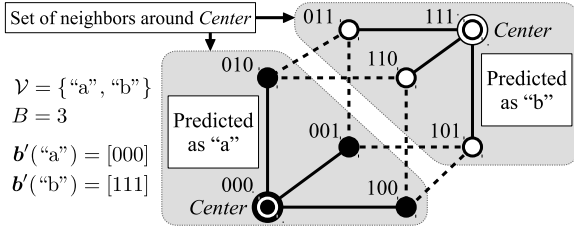

Figure 3: Example of the classification problem using redundant bit array mapping.

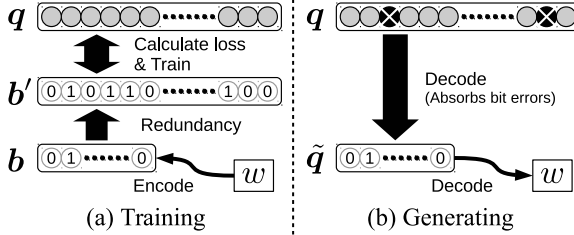

(a) Training ⋮ (b) Generating

Figure 4: Training and generation processes with error-correcting code.

max, because the actual size of the output layer is still small after applying the hybrid model.

### 3.5 Applying Error-correcting Codes

The 2 methods proposed in previous sections impose constraints for all bits in $q$, and the value of each bit must be estimated correctly for the correct word to be chosen. As a result, these models may generate incorrect words due to even a single bit error. This problem is the result of dense mapping between words and bit arrays, and can be avoided by creating *redundancy* in the bit array. Figure 3 shows a simple example of how this idea works when discriminating 2 words using 3 bits. In this case, the actual words are obtained by estimating the nearest *centroid* bit array according to the Hamming distance between each centroid and the predicted bit array. This approach can predict correct words as long as the predicted bit arrays are in the set of *neighbors* for the correct centroid (gray regions in the Figure 3), i.e., up to a 1-bit error in the predicted bits can be corrected. This ability to be robust to errors is a central idea behind *error-correcting codes* (Shannon, 1948). In general, an error-correcting code has the ability to correct up to $\lfloor (d-1)/2 \rfloor$ bit errors when all centroids differ $d$ bits from each other (Golay, 1949). $d$ is known as the *free distance* determined

by the design of error-correcting codes. Error-correcting codes have been examined in some previous work on multi-class classification tasks, and have reported advantages from the raw classification (Dietterich and Bakiri, 1995; Klautau et al., 2003; Liu, 2006; Kouzani and Nasireding, 2009; Kouzani, 2010; Ferng and Lin, 2011, 2013). In this study, we applied an error-correcting algorithm to the bit array obtained from Algorithm 1 to improve robustness of the output layer in an NMT system. A challenge in this study is trying a large classification (#classes > 10,000) with error-correction, unlike previous studies focused on solving comparatively small tasks (#classes < 100). And this study also tries to solve a generation task unlike previous studies. As shown in the experiments, we found that this approach is highly effective in these tasks.

Figure 4 (a) and (b) illustrate the training and generation processes for the model with error-correcting codes. In the training, we first convert the original bit arrays $\boldsymbol{b}(w)$ to a *center* bit array $\boldsymbol{b}'$ in the space of error-correcting code: $\boldsymbol{b}'(\boldsymbol{b}) := [b'_1(\boldsymbol{b}), b'_2(\boldsymbol{b}), \cdots, b'_{B'}(\boldsymbol{b})] \in \{0,1\}^{B'}$, where $B'(B) \geq B$ is the number of bits in the error-correcting code. The NMT model learns its parameters based on the loss between predicted probabilities $\boldsymbol{q}$ and $\boldsymbol{b}'$. Note that typical error-correcting codes satisfy $O(B'/B) = O(1)$, and this characteristic efficiently suppresses the increase of actual computation cost in the output layer due to the application of the error-correcting code. In the generation of actual words, the decoding method of the error-correcting code converts the redundant predicted bits $\boldsymbol{q}$ into a dense representation $\tilde{\boldsymbol{q}} := [\tilde{q}_1(\boldsymbol{q}), \tilde{q}_2(\boldsymbol{q}), \cdots, \tilde{q}_B(\boldsymbol{q})]$, and uses $\tilde{\boldsymbol{q}}$ as the bits to restore the word, as is done in the method described in the previous sections.

It should be noted that the method for performing error correction directly affects the quality of the whole NMT model. For example, the mapping shown in Figure 3 has only 3 bits and it is clear that these bits represent exactly the same information as each other. In this case, all bits can be estimated using exactly the same parameters, and we can not expect that we will benefit significantly from applying this redundant representation. Therefore, we need to choose an error correction method in which the characteristics of original bits should be distributed in various positions of the resulting bit arrays so that errors in bits are

---

**Algorithm 2** Encoding into a convolutional code.

---

**Require:** $b \in \{0,1\}^B$
**Ensure:** $b' \in \{0,1\}^{2(B+6)}$ = Redundant bit array

$$x[t] := \begin{cases} b_t, & \text{if } 1 \leq t \leq B \\ 0, & \text{otherwise} \end{cases}$$

$y_t^1 := x[t-6 \mathbin{..} t] \cdot [1001111] \mod 2$
$y_t^2 := x[t-6 \mathbin{..} t] \cdot [1101101] \mod 2$
$b' \leftarrow [y_1^1, y_1^2, y_2^1, y_2^2, \cdots, y_{B+6}^1, y_{B+6}^2]$

---

**Algorithm 3** Decoding from a convolutional code.

---

**Require:** $q \in (0,1)^{2(B+6)}$
**Ensure:** $\tilde{q} \in \{0,1\}^B$ = Restored original bit array

$g(q, b) := b \log q + (1-b) \log(1-q)$

$$\phi_0[s \mid s \in \{0,1\}^6] \leftarrow \begin{cases} 0, & \text{if } s = [000000] \\ -\infty, & \text{otherwise} \end{cases}$$

**for** $t = 1 \rightarrow B + 6$ **do**
 **for** $s^{\text{cur}} \in \{0,1\}^6$ **do**
 $s^{\text{prev}}(x) := [x] \circ s^{\text{cur}}[1 \mathbin{..} 5]$
 $o_1(x) := ([x] \circ s^{\text{cur}}) \cdot [1001111] \mod 2$
 $o_2(x) := ([x] \circ s^{\text{cur}}) \cdot [1101101] \mod 2$
 $g'(x) := g(q_{2t-1}, o_1(x)) + g(q_{2t}, o_2(x))$
 $\phi'(x) := \phi_{t-1}[s^{\text{prev}}(x)] + g'(x)$
 $\hat{x} \leftarrow \arg\max_{x \in \{0,1\}} \phi'(x)$
 $r_t[s^{\text{cur}}] \leftarrow s^{\text{prev}}(\hat{x})$
 $\phi_t[s^{\text{cur}}] \leftarrow \phi'(\hat{x})$
 **end for**
**end for**
$s' \leftarrow [000000]$
**for** $t = B \rightarrow 1$ **do**
 $s' \leftarrow r_{t+6}[s']$
 $\tilde{q}_t \leftarrow s'_1$
**end for**
$\tilde{q} \leftarrow [\tilde{q}_1, \tilde{q}_2, \cdots, \tilde{q}_B]$

---

not highly correlated with each-other. In addition, it is desirable that the decoding method of the applied error-correcting code can directly utilize the probabilities of each bit, because $q$ generated by the network will be a continuous probabilities between zero and one.

In this study, we applied convolutional codes (Viterbi, 1967) to convert between original and redundant bits. Convolutional codes perform a set of bit-wise convolutions between original bits and weight bits (which are hyper-parameters). They are well-suited to our setting here because they distribute the information of original bits in different places in the resulting bits, work robustly for random bit errors, and can be decoded using bit probabilities directly.

Algorithm 2 describes the particular convolutional code that we applied in this study, with two convolution weights [1001111] and [1101101] as fixed hyper-parameters.[2] Where $x[i \mathbin{..} j] :=$

---
[2] We also examined many configurations of convolutional codes which have different robustness and computation costs, and finally chose this one.

$[x_i, \cdots, x_j]$ and $x \cdot y := \sum_i x_i y_i$. On the other hand, there are various algorithms to decode convolutional codes with the same format which are based on different criteria. In this study, we use the decoding method described in Algorithm 3, where $x \circ y$ represents the concatenation of vectors $x$ and $y$. This method is based on the Viterbi algorithm (Viterbi, 1967) and estimates original bits by directly using probability of redundant bits. Although Algorithm 3 looks complicated, this algorithm can be performed efficiently on CPUs at test time, and is not necessary at training time when we are simply performing calculation of Equation (6). Algorithm 2 increases the number of bits from $B$ into $B' = 2(B+6)$, but does not restrict the actual value of $B$.

## 4 Experiments

### 4.1 Experimental Settings

We examined the performance of the proposed methods on two English-Japanese bidirectional translation tasks which have different translation difficulties: ASPEC (Nakazawa et al., 2016) and BTEC (Takezawa, 1999). Table 1 describes details of two corpora. To prepare inputs for training, we used `tokenizer.perl` in Moses (Koehn et al., 2007) and KyTea (Neubig et al., 2011) for English/Japanese tokenizations respectively, applied `lowercase.perl` from Moses, and replaced out-of-vocabulary words such that $\text{rank}(w) > V - 3$ into the `UNK` marker.

We implemented each NMT model using C++ in the DyNet framework (Neubig et al., 2017) and trained/tested on 1 GPU (GeForce GTX TITAN X). Each test is also performed on CPUs to compare its processing time. We used the *concat* local attention model proposed in Luong et al. (2015) constructed using a 1-layer LSTM (input/forget/output gates and non-peepholes) (Gers et al., 2000) encoder/decoder with 30% dropout (Srivastava et al., 2014) for the input/output vectors of the LSTMs. Only output layers and loss functions are replaced, and other network architectures are identical for the conventional/proposed models. We used the Adam optimizer (Kingma and Ba, 2014) with fixed hyper-parameters $\alpha = 0.001, \beta_1 = 0.9 \, \beta_2 = 0.999, \varepsilon = 10^{-8}$, and mini-batches with 64 sentences sorted according to their sequence lengths. For evaluating the quality of each model, we calculated BLEU (Papineni et al., 2002) every 1000 mini-batches. Table 2 lists all

Table 1: Details of the corpus.

| Name | | ASPEC | BTEC |
|---|---|---|---|
| Languages | | En ↔ Ja | |
| #sentences | Train | 2.00 M | 465. k |
| | Dev | 1,790 | 510 |
| | Test | 1,812 | 508 |
| Vocabulary size $V$ | | 65536 | 25000 |

Table 2: Evaluated models.

| Name | Summary |
|---|---|
| *Softmax* | Softmax prediction (Figure 2(a)) |
| *Binary* | Figure 2(b) w/ raw bit array |
| *Hybrid-N* | Figure 2(c) w/ softmax size $N$ |
| *Binary-EC* | *Binary* w/ error-correction |
| *Hybrid-N-EC* | *Hybrid-N* w/ error-correction |

models we examined in experiments.

## 4.2 Results and Discussion

Table 3 shows the BLEU on the test set, number of bits $B$ (or $B'$) for the binary code, actual size of the output layer $\#_{out}$, number of parameters in the output layer $\#_{W,\beta}$, as well as the ratio of $\#_{W,\beta}$ or amount of whole parameters compared with *Softmax*, and averaged processing time at training (per mini-batch on GPUs) and test (per sentence on GPUs/CPUs), respectively. Figure 5(a) and 5(b) shows training curves up to 180,000 epochs about some English→Japanese settings. To relax instabilities of translation qualities while training (as shown in Figure 5(a) and 5(b)), each BLEU in Table 3 is calculated by averaging actual test BLEU of 5 consecutive results around the epoch that has the highest dev BLEU.

First, we can see that each proposed method largely suppresses the actual size of the output layer from ten to one thousand times compared with the standard softmax. By looking at the total number of parameters, we can see that the proposed models require only 70% of the actual memory, and the proposed model reduces the total number of parameters for the output layers to a practically negligible level. Note that most of remaining parameters are used for the embedding lookup at the input layer in both encoder/decoder. These still occupy $O(EV)$ memory, where $E$ represents the size of each embedding layer and usually $O(E/H) = O(1)$. These are not targets to be reduced in this study because these values rarely are accessed at test time because we only need to access them for input words, and do not need them to always be in the physical memory. It might be

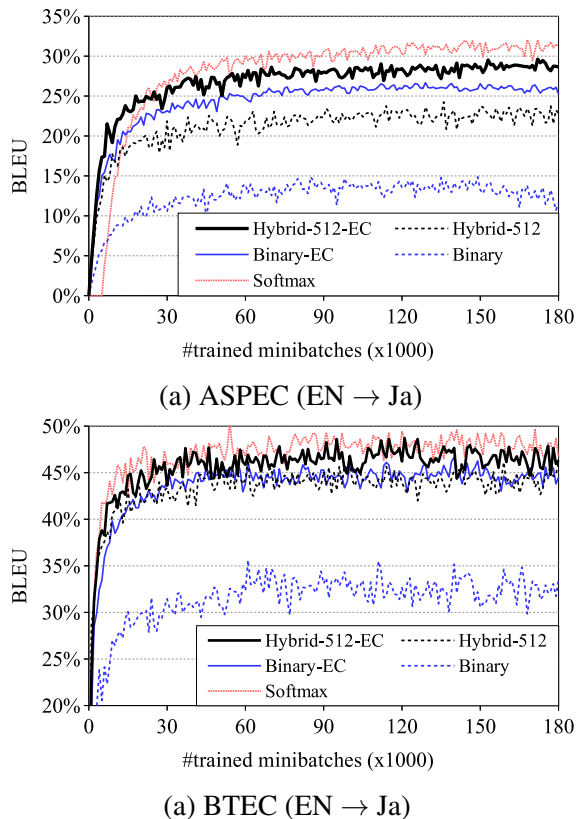

(a) ASPEC (EN → Ja)

(a) BTEC (EN → Ja)

Figure 5: Training curves over 180,000 epochs.

possible to apply a similar binary representation as that of output layers to the input layers as well, then express the word embedding by multiplying this binary vector by a word embedding matrix. This is one potential avenue of future work.

Taking a look at the BLEU for the simple *Binary* method, we can see that it is far lower than other models for all tasks. This is expected, as described in Section 3, because using raw bit arrays causes many one-off estimation errors at the output layer due to the lack of robustness of the output representation. In contrast, *Hybrid-N* and *Binary-EC* models clearly improve BLEU from *Binary*, and they approach that of *Softmax*. This demonstrates that these two methods effectively improve the robustness of binary code prediction models. Especially, *Binary-EC* generally achieves higher quality than *Hybrid-512* despite the fact that it suppress the number of parameters by about 1/10. These results show that introducing redundancy to target bit arrays is more effective than incremental prediction. In addition, the *Hybrid-N-EC* model achieves the highest BLEU in all proposed methods, and in particular, comparative or higher BLEU than *Softmax* in BTEC. This behav-

Table 3: Comparison of BLEU, size of output layers, number of parameters and processing time.

| Corpus | Method | BLEU % EnJa | BLEU % JaEn | $B$ | $\#_{out}$ | $\#_{W,\beta}$ | Ratio of #params $\frac{\#_{W,\beta}}{\#_{W,\beta}}$ | Ratio of #params All | Time (En→Ja) [ms] Train | Time (En→Ja) [ms] Test: GPU / CPU |
|---|---|---|---|---|---|---|---|---|---|---|
| ASPEC | *Softmax* | 31.13 | 21.14 | — | 65536 | 33.6 M | 1/1 | 1 | 1026. | 121.6 / 2539. |
| | *Binary* | 13.78 | 6.953 | 16 | 16 | 8.21 k | 1/4.10 k | 0.698 | 711.2 | 73.08 / 122.3 |
| | *Hybrid-512* | 22.81 | 13.95 | 16 | 528 | 271. k | 1/124. | 0.700 | 843.6 | 81.28 / 127.5 |
| | *Hybrid-2048* | 27.73 | 16.92 | 16 | 2064 | 1.06 M | 1/31.8 | 0.707 | 837.1 | 82.28 / 159.3 |
| | *Binary-EC* | 25.95 | 18.02 | 44 | 44 | 22.6 k | 1/1.49 k | 0.698 | 712.0 | 78.75 / 164.0 |
| | *Hybrid-512-EC* | 29.07 | 18.66 | 44 | 556 | 285. k | 1/118. | 0.700 | 850.3 | 80.30 / 180.2 |
| | *Hybrid-2048-EC* | 30.05 | 19.66 | 44 | 2092 | 1.07 M | 1/31.4 | 0.707 | 851.6 | 77.83 / 201.3 |
| BTEC | *Softmax* | 47.72 | 45.22 | — | 25000 | 12.8 M | 1/1 | 1 | 325.0 | 34.35 / 323.3 |
| | *Binary* | 31.83 | 31.90 | 15 | 15 | 7.70 k | 1/1.67 k | 0.738 | 250.7 | 27.98 / 54.62 |
| | *Hybrid-512* | 44.23 | 43.50 | 15 | 527 | 270. k | 1/47.4 | 0.743 | 300.7 | 28.83 / 66.13 |
| | *Hybrid-2048* | 46.13 | 45.76 | 15 | 2063 | 1.06 M | 1/12.1 | 0.759 | 307.7 | 28.25 / 67.40 |
| | *Binary-EC* | 44.48 | 41.21 | 42 | 42 | 21.5 k | 1/595. | 0.738 | 255.6 | 28.02 / 69.76 |
| | *Hybrid-512-EC* | 47.20 | 46.52 | 42 | 554 | 284. k | 1/45.1 | 0.744 | 307.8 | 28.44 / 56.98 |
| | *Hybrid-2048-EC* | 48.17 | 46.58 | 42 | 2090 | 1.07 M | 1/12.0 | 0.760 | 311.0 | 28.47 / 69.44 |

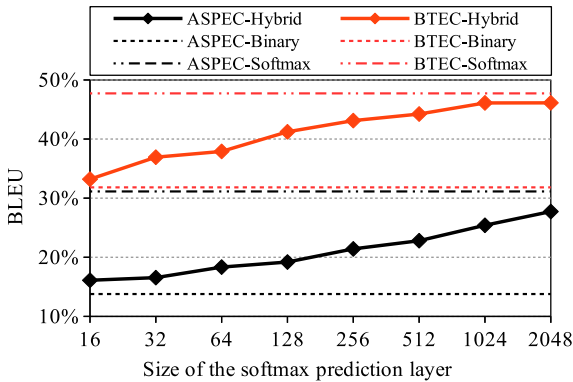

Figure 6: BLEU changes in the *Hybrid-N* methods according to the softmax size (En→Ja).

ior clearly demonstrates that these two methods are orthogonal, and combining them together can be effective. We hypothesize that the lower quality of *Softmax* in BTEC is caused by an over-fitting due to the large number of parameters required in the softmax prediction.

Proposed methods also improve actual computation time in both training and test. In particular, proposed methods can be performed significantly faster than *Softmax* at testing on CPUs by x5 to x20, which are directly affected by the complexity of output layers. In addition, we can also see that applying error-correcting code is also efficient at the point of the decoding speed.

Figure 6 shows the trade-off between the translation quality and the size of softmax layers in the hybrid prediction model (Figure 2(c)) without error-correction. According to the model definition in Section 3.4, the softmax prediction and

raw binary code prediction can be assumed to be the upper/lower-bound of the hybrid prediction model. The curves in Figure 6 move between *Softmax* and *Binary* models, and this behavior intuitively explains the characteristics of the hybrid prediction. In addition, we can see that the BLEU score in BTEC quickly improves, and saturates at $N = 1024$ in contrast to the ASPEC model, which is still improving at $N = 2048$. We presume that the shape of curves in Figure 6 is also affected by the difficulty of the corpus, i.e., when we train the hybrid model for easy datasets (e.g., BTEC is easier than ASPEC), it is enough to use a small softmax layer (e.g. $N \leq 1024$).

## 5 Conclusion

In this study, we proposed neural machine translation models which indirectly predict output words via binary codes, and two model improvements: a hybrid prediction model using both softmax and binary codes, and introducing error-correcting codes to introduce robustness of binary code prediction. Experiments show that the proposed model can achieve comparative translation qualities to standard softmax prediction, while significantly suppressing the amount of parameters in the output layer, and improving calculation speeds while training and especially testing.

One interesting avenue of future work is to automatically learn encodings and error correcting codes that are well-suited for the type of binary code prediction we are performing here. In Algorithms 2 and 3 we use convolutions that were determined heuristically, and it is likely that learning these along with the model could result in improved accuracy or better compression capability.

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
