# Peer review of "Neural Machine Translation via Binary Code Prediction"

_ACL 2017 — decision unknown_

[Official Review · Reviewer 1 · rating 4 · confidence 4]
soundness 3 · originality 4 · clarity 4 · impact 3 · substance 4 · appropriateness 5 · meaningful comparison 5 · presentation format Oral Presentation

- Strengths:
The proposed methods can save memory and improve decoding speed on CPUs without
losing (or a little loss) performance. 

- Weaknesses:
Since the determination of the convolutional codes of Algorithm 2 and Algorithm
3 can affect the final performance, I think it would be better if the authors
can explore a good method for it. And I think the argument of “Experiments
show the proposed model achieves translation accuracies that approach the
softmax, while reducing memory usage on the order of 1/10 to 1/1000, and also
improving decoding speed on CPUs by x5 to x20.” in the Abstract is not
rigorous. As far as I know, your experiments setting with “Binary” and
“Hybrid-512” on ASPEC corpus show the improvements of decoding speed on
CPUs by x20, but the BLEU scores are too low. So this is not a valid
conclusion.

- General Discussion:
This paper proposes an efficient prediction method for neural machine
translation, which predicts a binary code for each word, to reduce the
complexity of prediction. The authors also proposed to use the improved (error
correction) binary codes method to improve the prediction accuracy and the
hybrid softmax/binary model to balance the prediction accuracy and efficiency.
The proposed methods can save memory and improve decoding speed without losing
(or a little loss) performance. I think this is a good paper.

[Official Review · Reviewer 2 · rating 4 · confidence 5]
soundness 3 · originality 4 · clarity 4 · impact 3 · substance 4 · appropriateness 5 · meaningful comparison 5 · presentation format Poster

- Strengths:
  This paper is well written, and with clear, well-designed
  figures. The reader can easily understand the methodology even only
  with those figures.

  Predicting the binary code directly is a clever way to reduce the
  parameter space, and the error-correction code just works
  surprisingly well. I am really surprised by how 44 bits can achieve
  26 out of 31 BLEU.  
  The parameter reducing technique described in this work is
  orthogonal to current existing methods: weight pruning and
  sequence-level knowledge distilling.

  The method here is not restricted by Neural Machine Translation, and
  can be used in other tasks as long as there is a big output
  vocabulary.  

- Weaknesses:
  The most annoying point to me is that in the relatively large
  dataset (ASPEC), the best proposed model is still 1 BLEU point lower
  than the softmax model. What about some even larger dataset, like
  the French-English? There are at most 12 million sentences
  there. Will the gap be even larger?

  Similarly, what's the performance on some other language pairs ?

  Maybe you should mention this paper,
  https://arxiv.org/abs/1610.00072. It speeds up the decoding speed by
  10x and the BLEU loss is less than 0.5.  

- General Discussion:

The paper describes a parameter reducing method for large vocabulary
softmax. By applying the error-corrected code and hybrid with softmax,
its BLEU approaches that of the orignal full vocab softmax model.

One quick question: what is the hidden dimension size of the models?
I couldn't find this in the experiment setup.

The 44 bits can achieve 26 out of 31 BLEU on E2J, that was
surprisingly good. However, how could you increase the number of bits
to increase the classification power ? 44 is too small, there's plenty
of room to use more bits and the computation time on GPU won't even
change.

Another thing that is counter-intuitive is that by predicting the
binary code, the model is actually predicting the rank of the
words. So how should we interpret these bit-embeddings ? There seems
no semantic relations of all the words that have odd rank. Is it
because the model is so powerful that it just remembers the data ?

[Official Review · Reviewer 3 · rating 4 · confidence 3]
soundness 3 · originality 4 · clarity 5 · impact 3 · substance 4 · appropriateness 5 · meaningful comparison 5 · presentation format Oral Presentation

- Strengths:
This paper has high originality, proposing a fundamentally different way of
predicting words from a vocabulary that is more efficient than a softmax layer
and has comparable performance on NMT. If successful, the approach could be
impactful because it speeds up prediction.

This paper is nice to read with great diagrams. it's very clearly presented --
I like cross-referencing the models with the diagrams in Table 2. Including
loss curves is appreciated.

- Weaknesses:
Though it may not be possible in the time remaining, it would be good to see a
comparison (i.e. BLEU scores) with previous related work like hierarchical
softmax and differentiated softmax.

The paper is lacking a linguistic perspective on the proposed method. Compared
to a softmax layer and hierarchical/differentiated softmax, is binary code
prediction a natural way to predict words? Is it more or less similar to how a
human might retrieve words from memory? Is there a theoretical reason to
believe that binary code based approaches should be more or less suited to the
task than softmax layers?

Though the paper promises faster training speeds in the introduction, Table 3
shows only modest (less than x2) speedups for training. Presumably this is
because much of the training iteration time is consumed by other parts of the
network. It would be useful to see the time needed for the output layer
computation only.

- General Discussion:
It would be nice if the survey of prior work in 2.2 explicitly related those
methods to the desiderata in the introduction (i.e. specify which they
satisfy).

Some kind of analysis of the qualitative strengths and weaknesses of the binary
code prediction would be welcome -- what kind of mistakes does the system make,
and how does this compare to standard softmax and/or hierarchical and
differentiated softmax?

LOW LEVEL COMMENTS

Equation 5: what's the difference between id(w) = id(w') and w = w' ?

335: consider defining GPGPU

Table 3: Highlight the best BLEU scores in bold

Equation 15: remind the reader that q is defined in equation 6 and b is a
function of w. I was confused by this at first because w and h appear on the
LHS but don't appear on the right, and I didn't know what b and q were.